# Castration delays epigenetic aging and feminizes DNA methylation at androgen-regulated loci

Victoria J Sugrue[1], Joseph Alan Zoller[2], Pritika Narayan[3], Ake T Lu[4], Oscar J Ortega-Recalde[1], Matthew J Grant[3], C Simon Bawden[5], Skye R Rudiger[5], Amin Haghani[4], Donna M Bond[1], Reuben R Hore[6], Michael Garratt[1], Karen E Sears[7], Nan Wang[8], Xiangdong William Yang[8,9], Russell G Snell[3], Timothy A Hore[1†]*, Steve Horvath[4†]*

[1]Department of Anatomy, University of Otago, Dunedin, New Zealand; [2]Department of Biostatistics, Fielding School of Public Health, University of California, Los Angeles, Los Angeles, United States; [3]Applied Translational Genetics Group, School of Biological Sciences, Centre for Brain Research, The University of Auckland, Auckland, New Zealand; [4]Department of Human Genetics, David Geffen School of Medicine, University of California, Los Angeles, Los Angeles, United States; [5]Livestock and Farming Systems, South Australian Research and Development Institute, Roseworthy, Australia; [6]Blackstone Hill Station, Becks, RD2, Omakau, New Zealand; [7]Department of Ecology and Evolutionary Biology, UCLA, Los Angeles, United States; [8]Center for Neurobehavioral Genetics, Semel Institute for Neuroscience and Human Behavior, University of California, Los Angeles (UCLA), Los Angeles, United States; [9]Department of Psychiatry and Biobehavioral Sciences, David Geffen School of Medicine at UCLA, Los Angeles, United States

*For correspondence:
tim.hore@otago.ac.nz (TAH);
shorvath@mednet.ucla.edu (SH)

[†]These authors contributed equally to this work

**Abstract** In mammals, females generally live longer than males. Nevertheless, the mechanisms underpinning sex-dependent longevity are currently unclear. Epigenetic clocks are powerful biological biomarkers capable of precisely estimating chronological age and identifying novel factors influencing the aging rate using only DNA methylation data. In this study, we developed the first epigenetic clock for domesticated sheep (*Ovis aries*), which can predict chronological age with a median absolute error of 5.1 months. We have discovered that castrated male sheep have a decelerated aging rate compared to intact males, mediated at least in part by the removal of androgens. Furthermore, we identified several androgen-sensitive CpG dinucleotides that become progressively hypomethylated with age in intact males, but remain stable in castrated males and females. Comparable sex-specific methylation differences in *MKLN1* also exist in bat skin and a range of mouse tissues that have high androgen receptor expression, indicating that it may drive androgen-dependent hypomethylation in divergent mammalian species. In characterizing these sites, we identify biologically plausible mechanisms explaining how androgens drive male-accelerated aging.

## Introduction

Age has a profound effect on DNA methylation in many tissues and cell types (*Horvath, 2013*; *Issa, 2014*; *Rakyan et al., 2010*; *Teschendorff et al., 2010*). When highly correlated age-dependent sites are modeled through the use of a tool known as the epigenetic clock, exceptionally precise estimates of chronological age (termed 'DNAm age' or 'epigenetic age') can be achieved using only

purified DNA as an input (*Hannum et al., 2013*; *Horvath, 2013*; *Horvath and Raj, 2018*; *Levine et al., 2018*). For example, despite being one of the earliest epigenetic clocks constructed, Horvath's 353 CpG site clock is capable of estimating chronological age with a median absolute error (MAE) of 3.6 years and an age correlation of 0.96, irrespective of tissue or cell type (*Horvath, 2013*). Estimates generated by this and related epigenetic clocks are not only predictive of chronological age but also biological age, allowing identification of pathologies as well as novel genetic and environmental factors that accelerate or slow biological aging. For example, irrespective of ethnic background, females and exceptionally long-lived individuals are found to have reduced epigenetic aging compared to males and other controls (*Horvath et al., 2016*; *Horvath et al., 2015*).

Lifespan in mammals (including humans) is highly dependent upon an individual's sex, whereby females generally possess a longevity advantage over males (*Lemaître et al., 2020*). Despite being a fundamental risk factor affecting age-related pathologies, the mechanistic basis of how sex influences aging is relatively unexplored. Perhaps not surprisingly, sex hormones are predicted to play a central role, with both androgens and estrogens thought to influence aspects of the aging process (*Horstman et al., 2012*). Castration has been shown to extend the lifespan of laboratory rodents (*Asdell et al., 1967*), as well as domesticated cats (*Hamilton, 1965*) and dogs (*Hoffman et al., 2013*). Castration has also been associated with longer lifespans in historical survival reports of 14th- to 20th-century Korean eunuchs (*Min et al., 2012*) and men housed in US mental institutions in the 20th century (*Hamilton and Mestler, 1969*), although not in castrato opera singers, somewhat common in the 15th–19th centuries (*Nieschlag et al., 1993*). Conversely, estrogen production appears to have some protective effect on aging in females, with ovariectomized mice having a shortened lifespan (*Benedusi et al., 2015*) and replacement of ovaries from young animals into old female mice extending lifespan (*Cargill et al., 2003*). Indeed, ovariectomy has been shown to accelerate the epigenetic clock (*Stubbs et al., 2017*), supporting predictions that estrogen production slows the intrinsic rate of aging relative to males. In humans, natural and surgical induction of menopause also hastens the pace of the epigenetic clock, while menopausal hormone therapy decreases epigenetic aging as observed in buccal cell samples (*Levine et al., 2016*). Female breast tissue has substantially advanced epigenetic age compared to other tissues (*Horvath, 2013*; *Sehl et al., 2017*), further implicating sex hormones. Nevertheless, the effects of castration and/or testosterone production on the epigenetic predictors of aging in males have remained unknown in either humans or animal models prior to the current study.

Domesticated sheep (*Ovis aries*) represent a valuable, albeit underappreciated, large animal model for human disease and share with humans more similar anatomy, physiology, body size, genetics, and reproductive lifestyle as compared with commonly studied rodents (*Pinnapureddy et al., 2015*). With respect to aging, sheep exhibit a remarkable female-specific lifespan advantage (*Lemaître et al., 2020*), and Soay sheep of the Outer Hebrides represent a cornerstone research paradigm for longevity in wild mammal populations (*Fairlie et al., 2016*; *Jewell, 1997*). Moreover, sheep are extensively farmed (and males castrated) in many countries, allowing incidental study of the effect of sex and sex hormones in aging to occur without increasing experimental animal use (*Russell and Burch, 1959*). Yet, exploration of the molecular aging process in sheep is relatively underdeveloped, particularly from the perspective of epigenetics and sex.

Here, we present the first sheep epigenetic clock and quantify its median error to 5.1 months, ~3.5–4.2% of their expected lifespan. Validating the biological relevance of our sheep model, we find age-associated methylation at genes well characterized for their role in development and aging in a wide range of animal systems. Significantly, we observed that not only castration affects the epigenome, but that the methylomes of castrated male sheep show reduced epigenetic aging compared to intact male and female counterparts, a result consistent with the increased longevity of castrated Soay sheep (*Jewell, 1997*). Many genomic regions and genes with differential age association between castrated and intact males were identified, some of which are known to be regulated or bound by androgen receptor (AR) in humans and show sexually dimorphic methylation patterns in divergent mammalian species. Taken together, these findings provide a credible mechanistic link between levels of sex hormones and sex-dependent aging.

## Results

### DNA methylation in blood and ear throughout sheep aging

To create an epigenetic clock for sheep, we purified DNA from a total of 432 sheep of the Merino breed (*Figure 1A*). The majority of DNA samples (262) were from ear punches sourced from commercial farms in New Zealand, with the remaining (168) blood samples from a South Australian Merino flock. DNA methylation was quantified using a custom 38K probe array consisting of CpG sites conserved among a wide range of mammalian species; with 33,136 of these predicted to be complementary to sheep sequences. Two ear samples from intact males were excluded by quality control measures.

To initially characterize methylation data, we performed hierarchical clustering that revealed two major clusters based on tissue source (*Figure 1—figure supplement 1A*). There was some sub-clustering based on sex and age; however, there was no separation based on known underlying pedigree variation or processing batches. Global average CpG methylation levels in ear tissue exhibited a small progressive increase with age, though the same trend was not seen in blood (*Figure 1—figure supplement 1B*).

Pearson correlation coefficients (r) describing the linear relationship between CpG methylation and chronological age ranged from −0.63 to 0.68 for all ear and blood samples (*Figure 1B*, *Supplementary file 1*). One of the most positively correlated mapped probes was located within the promoter of fibroblast growth factor 8 (*FGF8*) (*Figure 1C*), a well-described developmental growth factor (r = 0.64, p=$1.38E^{-51}$). Probes located within several other well-known transcription factors (TFs) were also among those most highly correlated with age (*PAX6*, r = 0.62, p=$2.71E^{-47}$; *PAX5*, r = 0.59, p=$5.75E^{-43}$; *HOXC4*, r = 0.59, p=$4.47E^{-43}$). Indeed, when we performed ontogeny analysis, we found that the top 500 CpGs positively correlated with age were enriched for transcription-related and DNA binding processes (*Supplementary file 2*), consistent with widespread transcriptional shifts during different life stages. When we stratified our data, we found considerable differences in age association with regard to tissue of origin (*Figure 1—figure supplement 2*); however, this is consistent with other clock studies (*Horvath, 2013*; *Issa, 2014*). Sex and castration status also produced group-specific age association hits; however, this was comparatively less than that for tissue, with many significantly associated sites being shared between females and males (blood, 45 shared sites) and females, intact males, and castrated females (ear, 84 shared sites) (*Figure 1—figure supplement 2B, C*). Interestingly, we also found a CpG (cg18266944) in the second intron of insulin-like growth factor 1 (*IGF1*) that becomes rapidly hypomethylated in ear following birth before leveling off post-adolescence (*Figure 1D*; r = −0.60, p=$7.43E^{-15}$). We considered this a particularly encouraging age-associated epigenetic signal given that IGF1 is a key determinant of growth and aging (*Junnila et al., 2013*; *Laron, 2001*).

### Construction of an epigenetic clock in sheep

We established epigenetic clocks from our sheep blood and ear methylation data, respectively, as well as a combined blood and ear clock (hereafter referred to as the *multi-tissue clock*) using a penalized regression model (elastic-net regression). In total, 185 CpG sites were included in the multi-tissue clock, which was shown to have a MAE of 5.1 months and an age correlation of 0.95 when calculated using a leave-one-out cross-validation (LOOCV) analysis (*Figure 2A*). Taking into account the expected lifespan of sheep in commercial flocks (10–12 years), the error of the multi-tissue clock is 3.5–4.2% of the lifespan – comparable to the human skin and blood clock at ~3.5% (*Horvath et al., 2018*) and the mouse multi-tissue clock at ~5% of expected lifespan (*Meer et al., 2018*; *Petkovich et al., 2017*; *Stubbs et al., 2017*; *Thompson et al., 2018*; *Wang et al., 2017*). When blood and ear clocks are constructed separately, the ear clock outperformed the blood clock in the LOOCV analysis (0.97 vs. 0.75 correlation, respectively, *Figure 2B, C*). While this may be related to blood being a more heterogeneous tissue than ear punch, fewer samples and an overrepresentation of young individuals are the likely drivers of this effect.

Two human and sheep dual-species clocks were also constructed, which mutually differ by way of age measurement. One estimates the chronological ages of sheep and human (in units of years), while the other estimates the relative age – a ratio of chronological age of an animal to the maximum known lifespan of its species (defined as 22.8 years and 122.5 years for sheep and human,

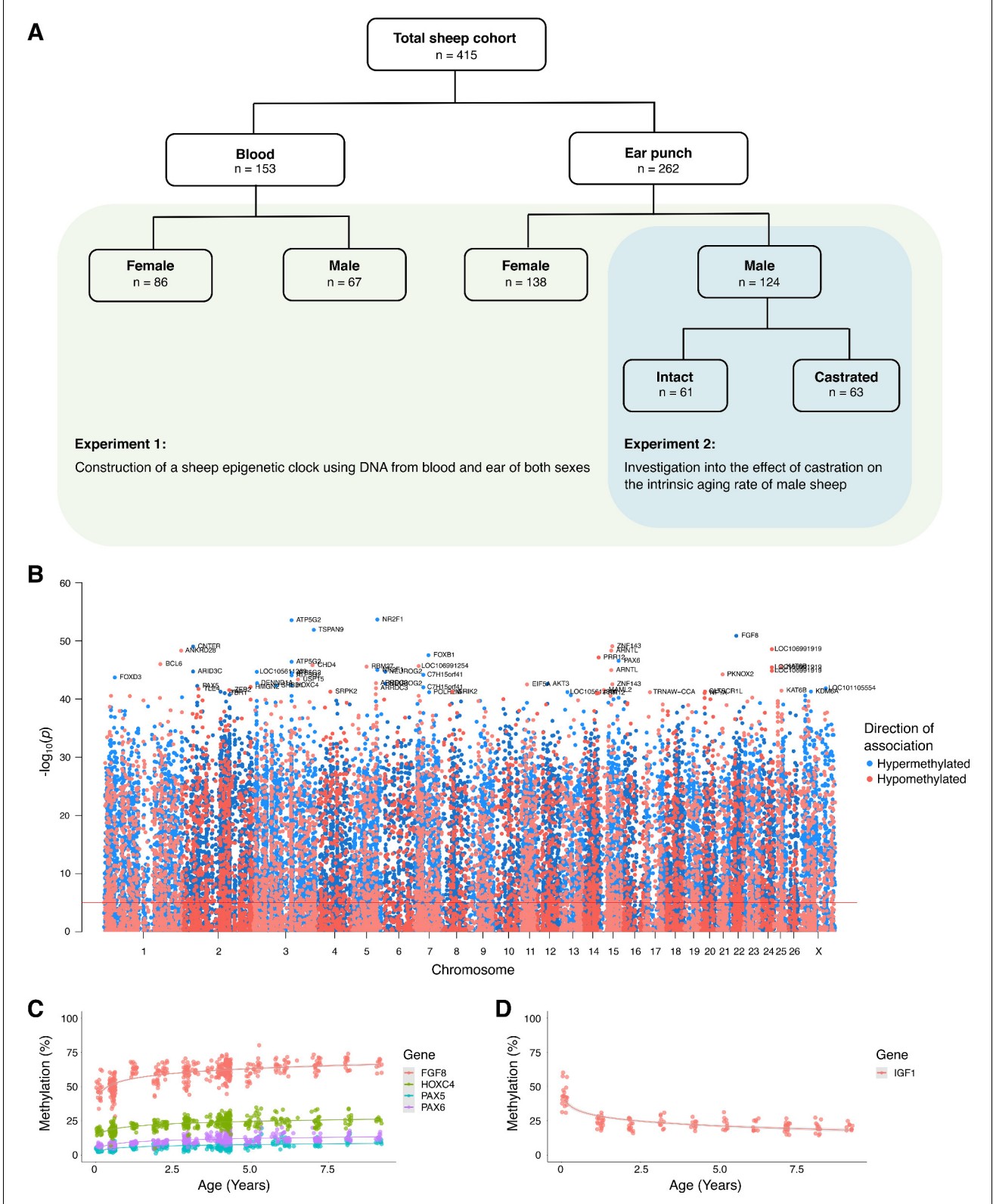

**Figure 1.** Association between age and DNA methylation in sheep. (**A**) Depiction of sheep cohort for this study. (**B**) Manhattan plot of all CpGs and their correlation with chronological age. (**C**) Methylation levels of highly age-correlated probes within biologically relevant genes: *FGF8* cg10708287 (r = 0.64, p=1.38E$^{-51}$), *PAX6* cg00953859 (r = 0.62, p=2.71E$^{-47}$), *PAX5* cg16071226 (r = 0.59, p=5.75E$^{-43}$), and *HOXC4* cg12097121 (r = 0.59,

*Figure 1 continued on next page*

*Figure 1 continued*

p=4.47E$^{-43}$). (D) Methylation levels of *IGF1* cg18266944 in ear of females only (r = 0.60, p=7.43$^{-15}$). The p-values of the correlation were calculated using the *standardScreeningNumericTrait* function in WGNCA (Student's t-test).

The online version of this article includes the following figure supplement(s) for figure 1:

**Figure supplement 1.** Characteristics of the sheep study cohort.
**Figure supplement 2.** Correlation between methylation and age with group stratification.

respectively) with resulting values between 0 and 1. The measure of relative age is advantageous as it aligns the ages of human and sheep to the same scale, yielding biologically meaningful comparison between the two species. The dual-species clock for chronological age leads to a median error of 16.53 months when considering both species or 4.74 months for sheep only (*Figure 2D, E*). The dual-species clock for relative age produced median errors of 0.020 of the maximum lifespans for both species (approximately 2.45 years for human or 5.4 months for sheep) and 0.021 for sheep only (approximately 5.7 months; (*Figure 2F, G*)).

## Castration delays epigenetic aging in sheep

To test the role of androgens in epigenetic age acceleration, we exploited the fact that castrated male Merino sheep are frequently left to age 'naturally' on New Zealand high-country farms in return for yearly wool production in contrast to non-breeding males of other sheep varieties, which are usually sold as yearlings for meat. Both castrated males and intact aged-matched controls were sourced from genetically similar flocks kept under comparable environmental conditions. Interestingly, intact and castrated males showed equivalent epigenetic age as juveniles; however, once they advanced beyond the yearling stage, castrates appeared to have slowed rates of epigenetic aging (*Figure 3A*). Indeed, when we only considered sheep beyond 18 months of age, we found that castrates had significantly reduced epigenetic age compared to intact male controls (*Figure 3B*, p=0.018). While the extent of age deceleration consistently increased with advancing age, mature castrates were on average epigenetically 3.1 months 'younger' than their chronological age (*Figure 3B*). In contrast, DNAm age of intact males was comparable to their chronological age (0.14 months age decelerated), as were females (0.76 months age accelerated), who comprised the majority of the samples from which the clock was constructed. Notably, the age deceleration observed in castrates was corroborated using the human and sheep dual-species clock (*Figure 3—figure supplement 1*, p=0.04).

To explore the mechanistic link between androgens and epigenetic aging, we identified 4694 probes with significant differences between the rate of age-dependent methylation changes in castrated or intact males (*Supplementary file 3*). A recent comparison of age-related methylation changes in the blood of human males and females revealed that almost all regions of interest appeared to be X-linked (*McCartney et al., 2019*). Given that there are already well-characterized differences between male and female methylation patterns on the X-chromosome as a result of gene-dosage correction (*Heard and Disteche, 2006*), it could be argued that X-linked age-related differences may be driven by peculiarities of methylation arising from X-chromosome inactivation as opposed to differences in androgen production per se. To test this for sheep, we examined the genomic location of our androgen-sensitive differentially methylated probes (asDMPs) and found that they are evenly distributed between individual autosomes and the sex chromosomes (*Figure 4—figure supplement 1A*).

Interestingly, we found several sites that become progressively hypomethylated in intact males with age but maintain a consistent level of methylation throughout life in castrates and females (*Figure 4A–D*). Indeed, of the top 50 most significantly different asDMPs, only 2 (cg03275335 *GAS1* and cg13296708 *TSHZ3*) exhibited gain of methylation in intact males (*Figure 5A*). We found that many asDMPs were linked to genes known to be regulated by AR (e.g., *MKLN1*, *LMO4*, *FN1*, *TIPARP*, *ZBTB16*; *Jin et al., 2013*), and as such, were encouraged to find further mechanistic connections between asDMPs and TF regulation.

To do this, we examined TF binding of the human regions homologous to our asDMPs using the Cistrome dataset; although Cistrome contains data from a wide range of TFs, we noted that AR binds to over half the top 50 asDMPs (28/50), with the 14 most significant all showing AR binding

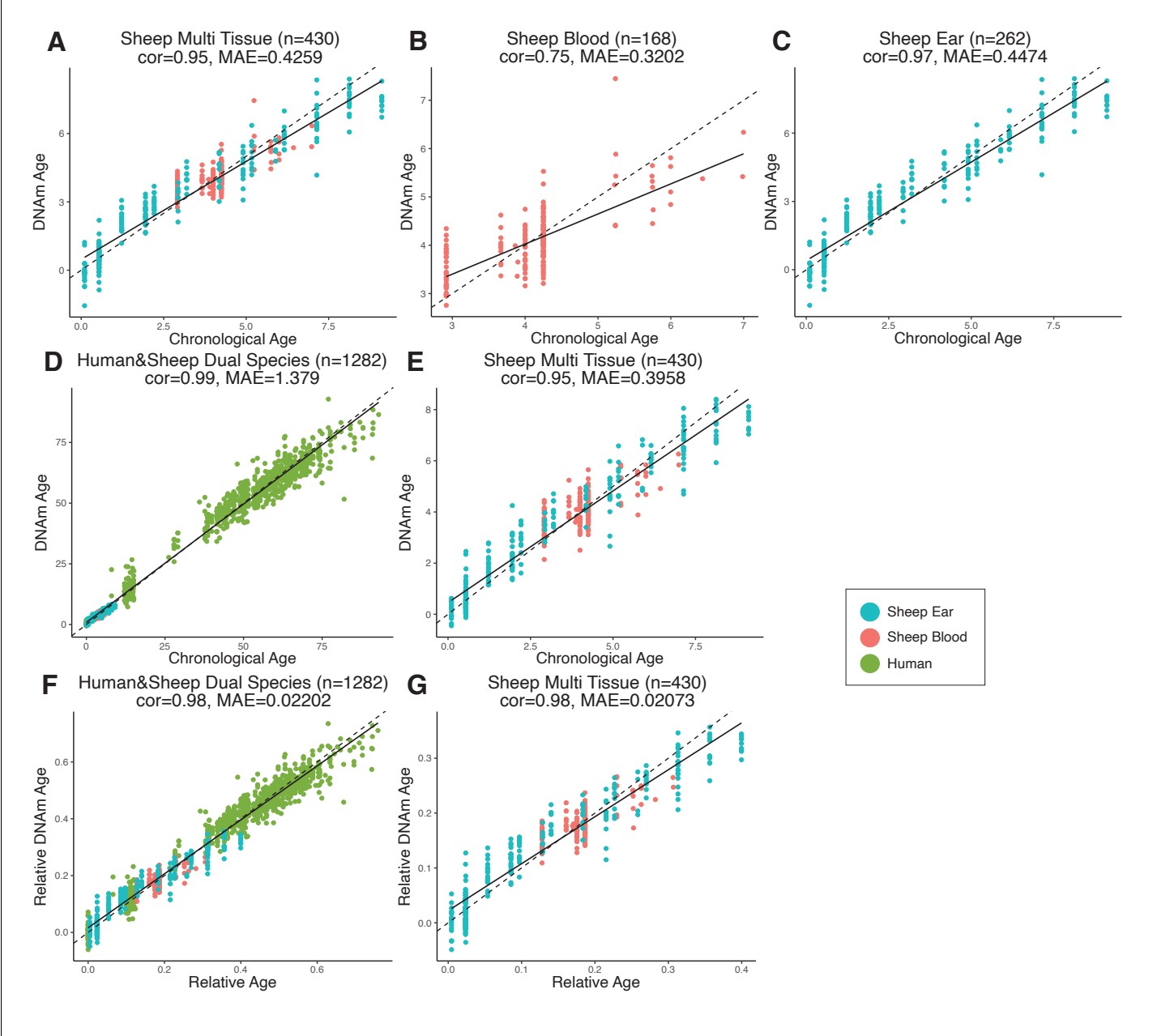

**Figure 2.** Comparison of chronological age (x-axis) and epigenetic age (y-axis) for a variety of clock models trained on (A–C) sheep only or (D–G) humans and sheep. Epigenetic age (DNAm age), correlation (cor), and median absolute error (MAE) are indicated for (A) sheep multi-tissue (ear and blood) (cor = 0.95, MAE = 0.4259), (B) sheep blood (cor = 0.75, MAE = 0.3202), and (C) sheep ear (cor = 0.97, MAE = 0.4474) clocks. For (D–G), clocks were created using methylation data from both humans and sheep, with DNAm age predictions displayed for (D) human and sheep (cor = 0.99, MAE = 1.379) and (E) sheep only (cor = 0.95, MAE = 0.3958) as calculated using absolute time (years). DNAm age was also calculated relative to maximum lifespan for (F) human and sheep (cor = 0.98, MAE = 0.02202) and (G) sheep only (cor = 0.98, MAE = 0.02073). Maximum lifespan values used were for human and sheep, respectively, were 122.5 years and 22.8 years. Each data point represents one sample, colored based on origin.

(*Figure 5A*). To ensure that this was not a result expected by chance alone, we performed empirical sampling whereby 1000 replicates of binding analysis were performed, but with 50 random CpG sites from the methylation array at each bootstrap replicate. The observed/expected enrichment for AR binding in the top 50 asDMPs within the Cistrome dataset was 9.9-fold (16.5%/1.67%, p<0.001), and along with the glucocorticoid receptor (GR) (NR3C1, 6.7-fold enrichment), was much greater than any other TF. Nevertheless, there were several other interesting related TFs with observed/

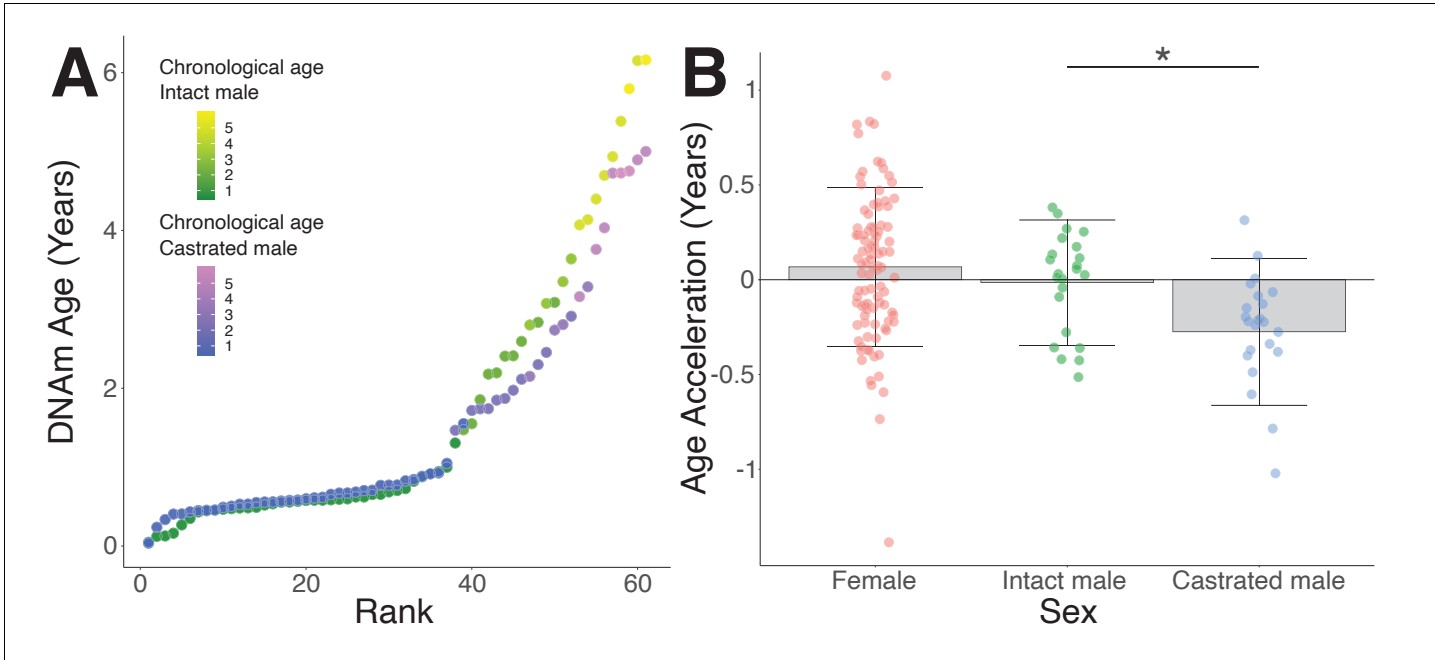

**Figure 3.** Epigenetic age deceleration in castrated sheep. (A) Epigenetic age in age-matched castrated and intact males. To equate the cohort sizes for intact and castrated males, two age-matched castrates with DNAm age estimates closest to the group mean were excluded. (B) Age acceleration based on sex and castration status in sexually mature sheep only (ages 18 months+ only). Castrated males have decelerated DNAm age compared to intact males (*p=0.018, Mann–Whitney U test).

The online version of this article includes the following figure supplement(s) for figure 3:

**Figure supplement 1.** Age acceleration based on sex and castration status in sexually mature sheep only (age 18 months+ only) using the human and sheep dual-species clock.

expected ratios >2 for the top 50 asDMPs, including the estrogen (ESR1) and progesterone (PR) receptors, as well as CCAAT/enhancer-binding protein beta (CEBP) and forkhead box A1 (FOXA1) (*Figure 5B*). These effects were not seen across all asDMPs - high observed/expected binding ratios were only found in the first 150 most significant asDMPs (e.g., AR and NR3C1 in *Figure 4—figure supplement 1C, D*), after which only background enrichment was seen.

While we saw similar features at other asDMPs (*Figure 5—figure supplement 1*), the asDMP that was the most different in epigenetic aging rate between castrates and males, *MKLN1*, stood out as being particularly interesting from a gene regulatory perspective (*Figure 5C*). Overlapping with this site, and AR binding, were peaks of DNase I hypersensitivity, H3K27ac histone marks, as well as good vertebrate conservation compared to surrounding sequences.

## asDMPs are associated with body mass in juvenile males and conserved in divergent mammalian species in a tissue-specific manner

Juvenile males in our study did not show alterations in epigenetic age according to castration status (*Figure 3A*). Nevertheless, mass of these animals was recorded, and intact males were on average heavier than castrates, implying that there are phenotypic consequences of androgen exposure even during early puberty (*Figure 6A*). To explore this further, we tested if there was any relationship between the mass of juvenile males and methylation at asDMPs. Interestingly, when examined in windows of 100 (thus reducing noise), the most prominent asDMPs showed a negative association with sheep mass in intact males, but a positive association in castrates (*Figure 4—figure supplement 2B*). As less prominent asDMPs were analyzed, the significant difference between the castrate and intact male associations was lost (*Figure 4—figure supplement 2C*). Together, this shows that divergence of methylation at asDMP sites occurs early (i.e., as soon as physical traits like mass become dimorphic between intact and castrated males) and before differences in epigenetic age are manifested.

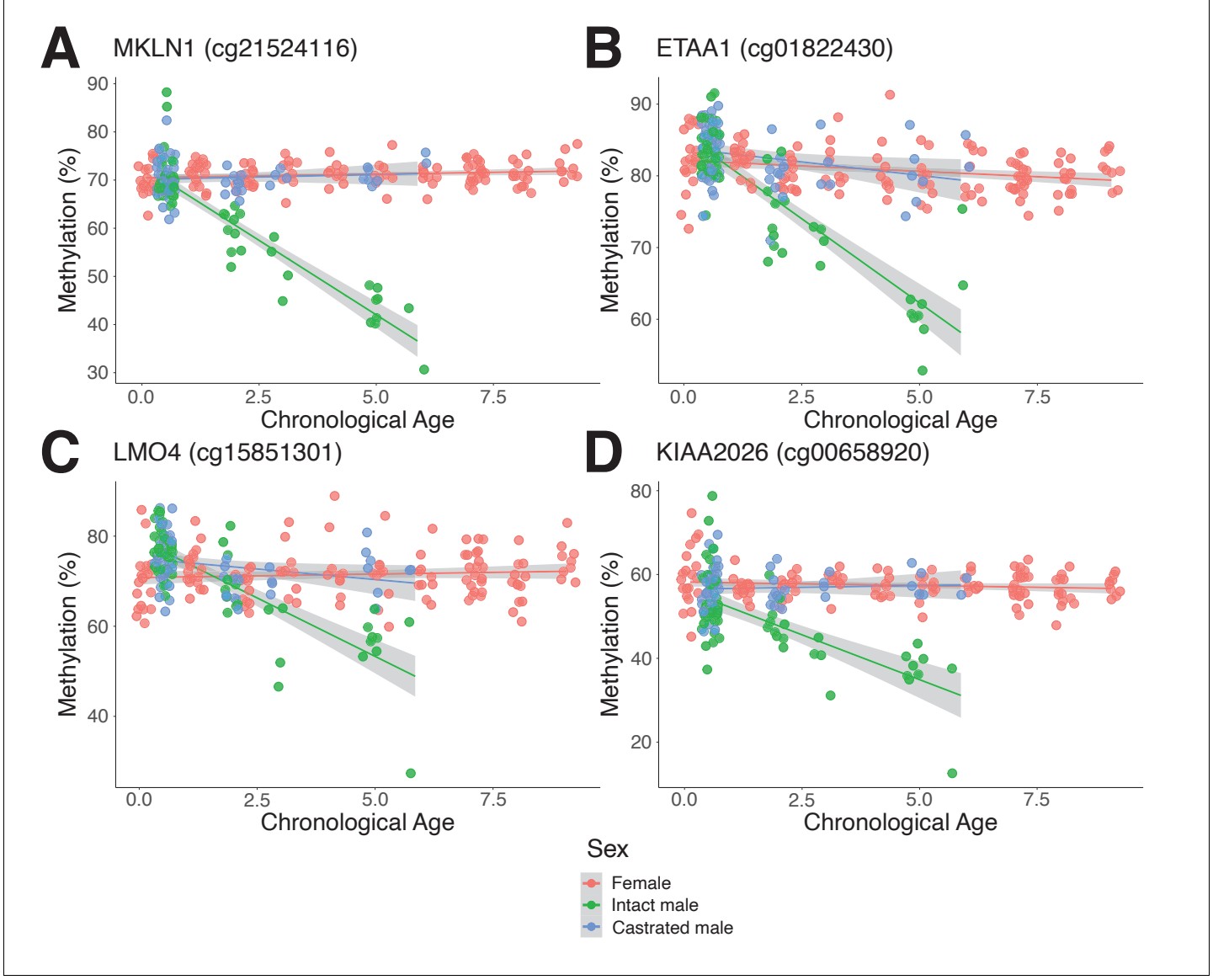

**Figure 4.** Androgen-sensitive differentially methylated probes (asDMPs) in sheep ear. (**A**) *MKLN1* (cg21524116, p=1.05E$^{-27}$), (**B**) *ETAA1* (cg01822430, p=1.31E$^{-13}$), (**C**) *LMO4* (cg15851301, p=1.62E$^{-09}$), and (**D**) *KIAA2026* (cg00658920, p=2.46E$^{-09}$). The p-values were calculated using a t-test of the difference in linear regression slopes.

The online version of this article includes the following figure supplement(s) for figure 4:

**Figure supplement 1.** Androgen-sensitive differentially methylated probes (asDMPs).

**Figure supplement 2.** Mass and methylation in young male sheep.

While the discovery of androgen-dependent age-associated methylation in sheep ear appeared striking, particularly for sites like MKLN1, we were unsure as to how widespread this phenomenon would be. Indeed, it appeared that significant male-specific hypomethylation was not present in sheep blood; and such compelling examples of this effect had not been reported previously in other species (most studies of which also involved blood). To explore this further, we assessed methylation changes in several mouse tissues using the pan-mammalian array. Again, cg21524116 in *MKLN1* stood out – muscle, tail, and kidney exhibited the same male-specific hypomethylation as seen in sheep, whereby females retain a constant level of high methylation (*Figure 6A, B*). Importantly, however, this same trend could not be seen in mouse blood, cerebellum, cortex, and liver. To explain this tissue specificity, we made use of microarray expression data from mouse (*Su et al., 2002*) and

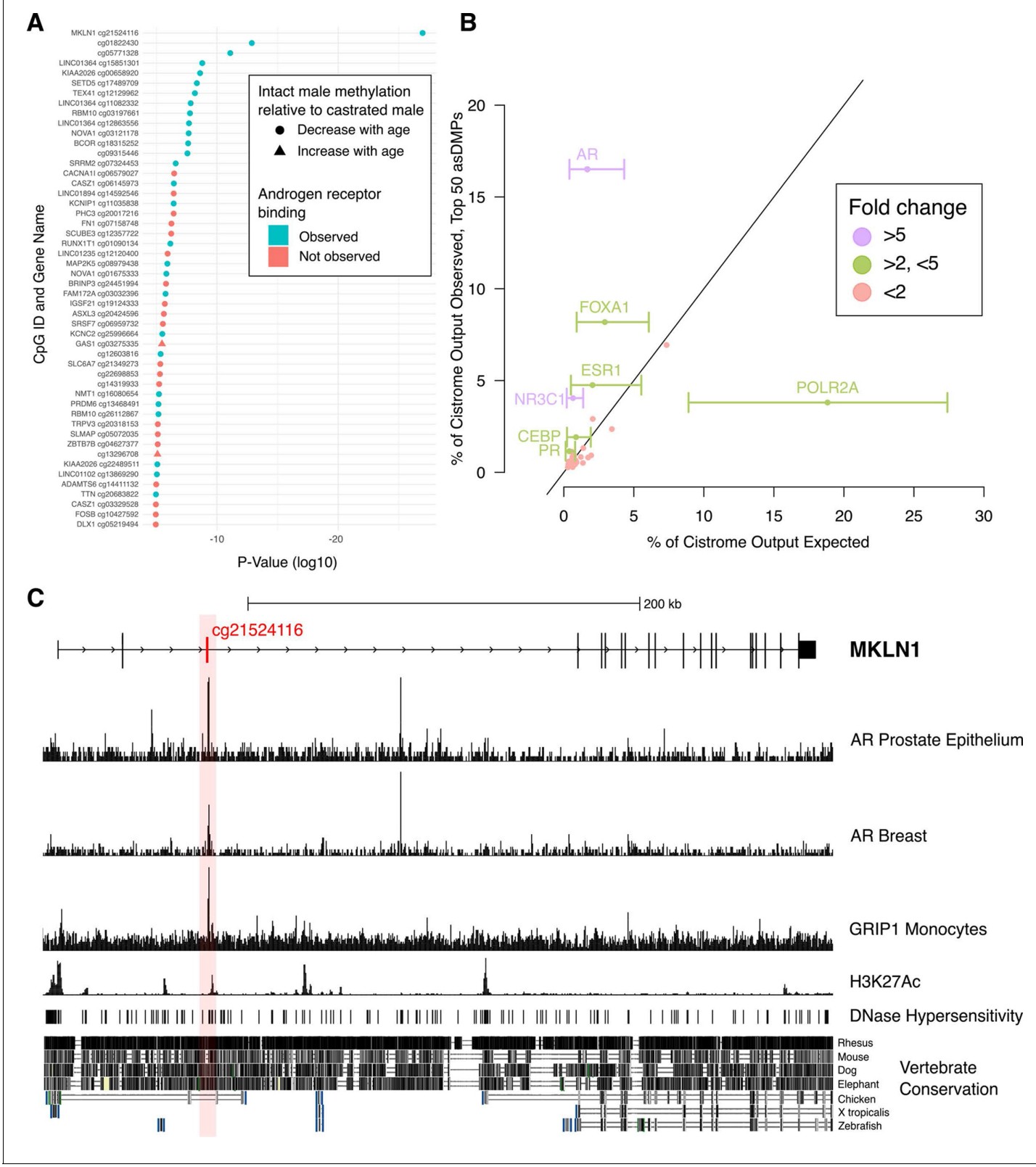

**Figure 5.** Analysis of chromatin immunoprecipitation and sequencing (ChIP-seq) data indicates functional links to sex-dependent epigenetic aging. (**A**) Top 50 androgen-sensitive differentially methylated probes (asDMPs) between intact and castrated male sheep, and the human genes they map to (where applicable). The top 14 most significant asDMPs are all bound by androgen receptor (AR); and 48/50 of these sites exhibit hypomethylation with age in intact males relative to castrated males. (**B**) Observed transcription factor (TF) binding at the top 50 asDMPs compared to expected binding

*Figure 5 continued on next page*

*Figure 5 continued*
based upon empirical sampling at random CpGs (average of 1000 bootstrap replicates). TFs with greater than twofold variation and an absolute value of >1% are labeled with error bars showing the range of TF binding in bootstrap sampling. Colors indicate fold-change between observed and expected TF binding; <2 (red), 2–5 (green), and >5 (purple) (C) Genomic view of *MKLN1* containing the most significant asDMP cg21524116 illustrating AR binding and indicators of active chromatin.

The online version of this article includes the following figure supplement(s) for figure 5:

**Figure supplement 1.** Gene views of key androgen-sensitive sites showing evidence for possible regulatory functions.

found that the AR is highly expressed in muscle, kidney, and (epidermis of) skin, tissues where androgen sensitivity at MKLN1 was also observed (*Figure 6C*). In contrast, AR expression is significantly lower in cerebellum, cortex, and liver where male-specific hypomethylation at MKLN1 is not observed.

Currently, few datasets using the pan-mammalian array have been published that involve skin or other tissues where AR expression is expected to be high. Nevertheless, DNA methylation levels in skin from a range of bat species have recently been released (*Wilkinson et al., 2021*). We found that at least within *Phyllostomus* and *Pteropus* genera, where sufficient numbers of male and female samples were collected, male-specific hypomethylation of MKLN1 was present (*Figure 6—figure supplement 1*). When considered alongside the mouse data, this implies that androgen-dependent

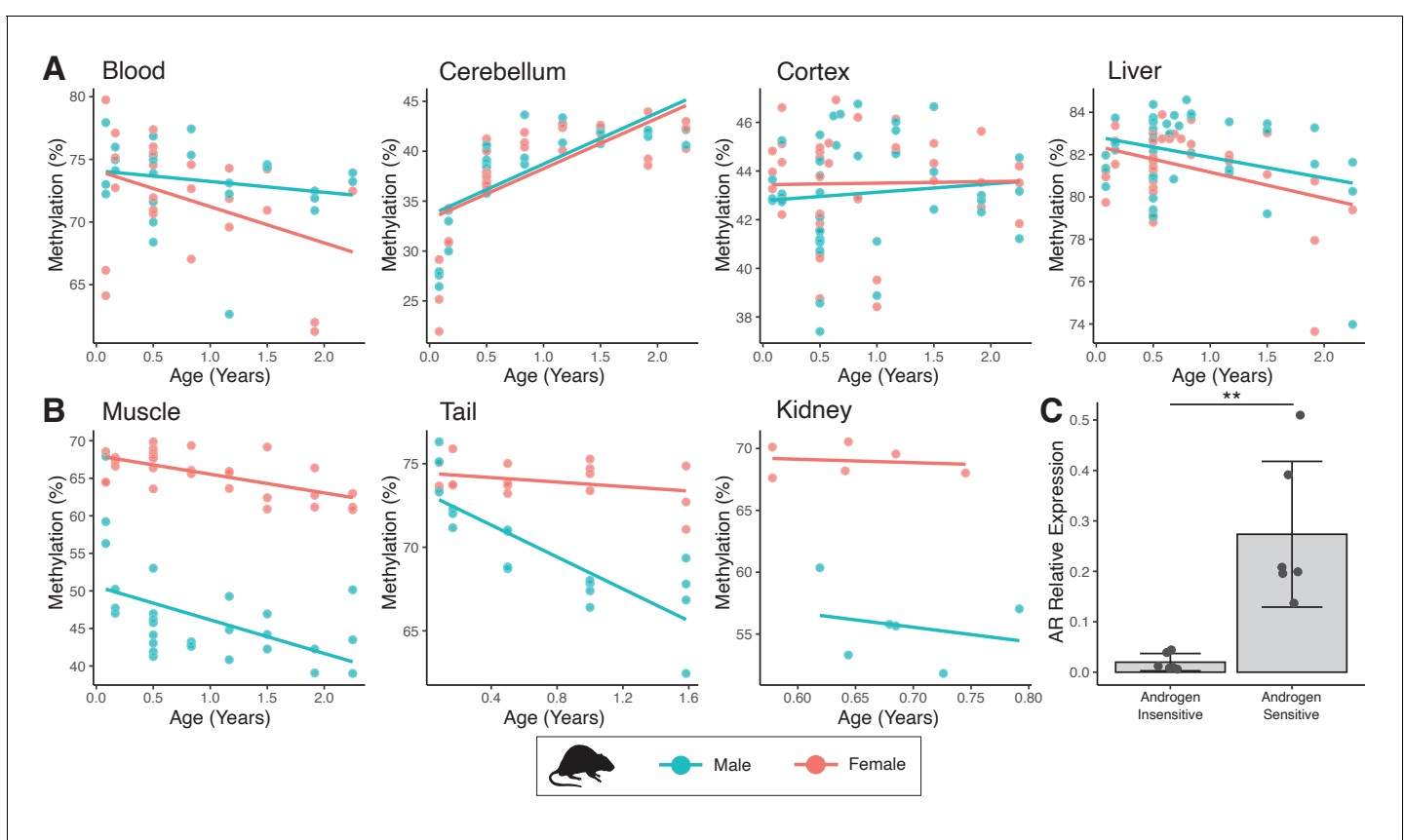

**Figure 6.** Sex-specific methylation patterns for cg21524116 (*MKLN1*) in mouse tissues are linked to androgen receptor (AR) expression. In specific tissues of mouse, probe cg21524116 (*MKLN1*) exhibits either (A) no sex differences or (B) male exaggerated hypomethylation during aging. (C) AR mRNA expression is significantly higher in tissues showing male hypomethylation at MKLN1 (epidermis, kidney, skeletal muscle) compared to tissues where MKLN1 methylation is not different between sexes (cerebellum, cortex, liver) (p=0.007; t-test).

The online version of this article includes the following figure supplement(s) for figure 6:

**Figure supplement 1.** Androgen-sensitive methylation patterns at cg21524116 (*MKLN1*) in skin of several bat species.

alterations in DNA methylation age association exist, at least to some extent, in a wide range of mammalian species.

## Discussion

Epigenetic clocks are accurate molecular biomarkers for aging that have proven to be useful for identifying novel age-related mechanisms, diseases, and interventions that alter the intrinsic aging rate (*Horvath and Raj, 2018*). Here, we developed the first epigenetic clock for sheep and show that it is capable of estimating chronological age with a MAE of 5.1 months – between 3.5% and 4.2% of the average sheep lifespan. Significantly, we also present the first evidence that castration feminizes parts of the epigenome and delays epigenetic aging.

Improved survival has previously been reported in castrated sheep compared to intact males and females, at least part of which has been attributed to behavioral changes such as reduced aggression (*Jewell, 1997*). Our data shows that castration also causes a delay in intrinsic aging as assessed by the epigenetic clock, with an average reduction in epigenetic age of 3.1 months (*Figure 3B*). Moreover, delayed epigenetic aging in castrates is also seen relative to intact males and females, which is consistent with castrated males outliving intact animals of both sexes (*Jewell, 1997*). We also find that the degree of age deceleration observed in castrated males is dependent on their chronological age. For instance, the average DNAm deceleration is increased by an additional 1.2 months when considering individuals aged beyond 2.9 years. In contrast, we saw no difference between castrated and intact males younger than 18 months. Together this implies that the effects of androgen exposure on the epigenome and aging are cumulative. Similar findings of greater age deceleration at later chronological ages have been observed in rodent models, with long-lived calorie-restricted mice showing a younger epigenetic age relatively late in life, but similar epigenetic aging rates at younger ages (*Petkovich et al., 2017*).

These results support the reproductive cell-cycle theory as an explanation for sex-dependent differences in longevity of mammals (*Atwood and Bowen, 2011*; *Bowen and Atwood, 2004*). Androgens and other testicular factors may be working in an antagonistic pleiotropic manner whereby they push cells through the cell cycle and promote growth in early life to reach reproductive maturity, thus also influencing the epigenome in an age-related manner. This process, however, may become dysregulated and promote senescence at older ages, reflected in the hastening of the epigenetic clock observed in intact males compared to castrates. It is well known in farming practice that where it can be managed appropriately, leaving male sheep intact or partially intact (i.e., cryptorchid) increases body mass (*Seideman et al., 1982*), something we also observed in our study (*Figure 4—figure supplement 2A*). This indicates greater rates of cell cycle progression, cellular division, and tissue hyperplasia. Under this hypothesis, the effects of castration should depend on whether animals are castrated before or after puberty. In rats, castration just after birth (i.e., prior to puberty) causes substantial lifespan extension while castration after puberty has smaller effects (*Talbert and Hamilton, 1965*), supporting the idea that male gonadal hormones have effects at early-life stages that have deleterious consequences for survival.

Consequences of castration for increased survival and slowed epigenetic aging could also be linked to the effects of androgens on sexual dimorphism and adult reproductive investment (*Brooks and Garratt, 2017*). Life history theories predict that males in highly polygynous species, like sheep, should be selected to invest heavily in reproduction early in life, even at the expense of a shorter lifespan, because they have the potential to monopolize groups of females and quickly produce many offspring (*Clutton-Brock and Isvaran, 2007*; *Tidière et al., 2015*). By contrast, selection on females should promote a slower reproductive life strategy because female reproductive rate is limited by the number of offspring they can produce. While we show that castration slows epigenetic aging in sheep, loss of ovarian hormone production in mice and human (through ovariectomy or menopause) is associated with a hastening of the epigenetic clock (*Levine et al., 2016*; *Stubbs et al., 2017*), consistent with the beneficial effects of female ovarian hormones on survival. Thus, it appears that both male and female sex hormones differentially regulate the epigenetic aging process in directly opposing ways, in a manner that is consistent with the life history strategies classically thought to be optimal for each sex.

Comparison of intact and castrated males also allowed us to identify several age-related DMPs that display clear androgen-sensitivity (asDMPs, *Figure 4A–D*), with castrated males exhibiting a

feminized methylation profile compared to intact counterparts at these sites. In contrast to similar experiments performed in human blood, we found that these sex-specific CpG sites are not predominantly X-linked in sheep (*McCartney et al., 2019*), but are instead distributed evenly throughout the genome (*Figure 4—figure supplement 1A*). As yet, we do not know if castration in later life would drive feminization of methylation patterns as we observed for early-life castration (*Figure 4*). This is, however, an interesting consideration – it is possible that castration late in life would quickly recapitulate the methylation differences seen in those castrated early in life or it may be that methylation patterns established during early growth and development are difficult to change once set on a particular aging trajectory. This distinction may be important from a functional perspective because early- and later-life castration can have differing effects on survival in rodents (*Talbert and Hamilton, 1965*). Moreover, while early-life castration has been shown to extend human lifespan, androgen depletion in elderly men can be associated with poor health (*Araujo et al., 2011*).

Many of the highly significant asDMPs (i.e., 28/50, or 58%) are bound by AR (*Figure 5A*). When considering TF binding compared to background levels, our data shows that particularly AR, but also NR3C1, ESR1, FOXA1, and CEBP binding are enriched in the most significant asDMPs; all of which share biologically integrated functions. NR3C1, which encodes the GR and has been previously linked to longevity in certain populations (*Olczak et al., 2019*), is an anabolic steroid receptor; thus, it shares significant homology in its binding domain with AR and targets many DNA sequences also bound by AR (*Claessens et al., 2017*). CEBP is known to directly bind AR in prostate cells (*Agoulnik et al., 2003*). FOXA1 has been found to regulate estrogen receptor binding (*Carroll et al., 2005*; *Hurtado et al., 2011*) as well as AR and GR binding (*Sahu et al., 2013*) in both normal and cancer cells. Furthermore, FOXA1 aids in ESR1-mediated recruitment of GRs to estrogen receptor binding regions (*Karmakar et al., 2013*). Interestingly, AR agonist treatment in breast cancer models reprograms binding of both FOXA1 and ESR (*Ponnusamy et al., 2019*), suggesting some degree of antagonistic function between the androgen and estrogen receptors. If this is true for asDMPs, these sites may well represent a conduit through which castrates and androgen-deprived males take on physiologically feminized traits, including delayed aging.

Having said this, it remains a possibility that methylation levels at androgen-sensitive sites have very little to do with biological aging and instead only diverge as time progresses due to the period of androgen exposure or deficiency. Specifically, the changes in methylation observed between intact and castrated males may not be adaptive at all, and rather, methylation is progressively 'diluted' by binding of AR to the DNA. Variable methylation at AR target genes has been reported in humans with androgen insensitivity syndrome (AIS) when compared to normal controls (*Ammerpohl et al., 2013*), supporting the notion that AR binding could influence age-associated methylation profiles.

The most striking example of age-dependent androgen-sensitive DNA methylation loss we found was that detected by the probe cg21524116, mapping to *MKLN1* (Muskelin) (*Figure 4A*). Evidence for *MKLN1* androgen-dependence has previously been reported (*Jin et al., 2013*), and MKLN1-containing complexes have been shown to regulate lifespan in *Caenorhabditis elegans* (*Hamilton et al., 2005*; *Liu et al., 2020*), although no links between this gene and mammalian longevity are yet known. Chromatin immunoprecipitation and sequencing (ChIP-seq) data demonstrates enriched AR binding at the position of this asDMP in human, as well as exhibiting high-sequence conservation, DNase hypersensitivity and H3K27ac marks – the latter two of which are markers of open chromatin and indicate transcriptionally active areas (*Figure 5C*; *Creyghton et al., 2010*; *Wang et al., 2008*). Significantly, we found that *MKLN1* hypomethylation can be seen in skin from male bats, as well as tail, muscle, and kidney from mice – all tissues where there is high expression of AR (*Figure 6A, B*). In contrast, tissues showing no sex-dependent methylation changes in *MKLN1* during aging (i.e., cortex, cerebellum, and liver) show lower or silenced expression of AR. While this needs to be tested in more species and tissues, together this implies that *MKLN1* and other asDMPs represent potential biomarkers of androgen exposure over long time periods in a broad range of mammals.

In summary, this paper describes a robust epigenetic clock for sheep that is capable of estimating chronological age, detecting accelerated rates of aging, and contributes to a growing body of work on epigenetic aging. In addition to demonstrating the utility of sheep as an excellent model for aging studies, our data identify androgen-dependent age-associated methylation changes that affect known targets of sex hormone pathways and hormone binding TFs. While these changes may not promote aging per se, identification of loci with age-dependent androgen-sensitive methylation

patterns uncovers novel mechanisms by which male-accelerated aging in mammals can be explained.

## Materials and methods

### DNA extraction and quantitation

Sheep DNA samples for this study were derived from two distinct tissues from two strains: ear tissue from New Zealand Merino and blood from South Australian Merino.

### Sheep ear DNA source

Ear tissue was obtained from females and both intact and castrated male Merino sheep during routine on-farm ear tagging procedures in Central Otago, New Zealand. As a small piece of tissue is removed during the ear tagging process that is usually discarded by the farmer, we were able to source tissue and record the year of birth without altering animal experience, in accordance with the New Zealand Animal Welfare Act (1999) and the National Animal Ethics Advisory Committee (NAEAC) Occasional Paper No 2 (*Carsons, 1998*). The exact date of birth for each sheep is unknown; however, this was estimated to be the 18th of October each year, according to the date at which rams were put out with ewes (May 10th of each year), a predicted mean latency until mating of 12 days, and the mean gestation period from a range of sheep breeds (149 days) (*Fogarty et al., 2005*). Castration was performed by the farmer using the rubber ring method within approximately 5–50 days from birth as per conventional farming practice (*National Animal Welfare Advisory Comittee NZ, 2018*). Mass of yearlings was recorded by the farmer for both castrated and intact male sheep at 6.5 months of age as a part of routine growth assessment. In total, ear tissue from 138 female sheep aged 1 month to 9.1 years and 126 male sheep (63 intact, 63 castrates) aged 6 months to 5.8 years was collected and subjected to DNA extraction (*Figure 1A*).

DNA was extracted from ear punch tissue using a Bio-On-Magnetic-Beads (BOMB) protocol (*Oberacker et al., 2019*), which isolates DNA molecules using solid-phase reversible immobilization (SPRI) beads. Approximately 3 mm punches of ear tissue were lysed in 200 µL TNES buffer (100 mM Tris, 25 mM NaCl, 10 mM EDTA, 10% w/v SDS), supplemented with 5 µL 20 mg/mL Proteinase K and 2 µL RNAse A and incubated overnight at 55°C as per BOMB protocols. The remainder of the protocol was appropriately scaled to maximize DNA output while maintaining the necessary 2:3:4 ratio of beads:lysate:isopropanol. As such, 40 µL cell lysate, 80 µL 1.5X GITC (guanidinium thiocyanate), 40 µL TE-diluted Sera-Mag Magnetic SpeedBeads (GE Healthcare, GEHE45152105050250), and 80 µL isopropanol were combined. After allowing DNA to bind the SPRI beads, tubes were placed on a neodymium magnetic rack for ~5 min until the solution clarified and supernatant was removed. Beads were washed 1× with isopropanol and 2× with 70% ethanol, and then left to air dry on the magnetic rack. 25 µL of MilliQ H$_2$O was added to resuspend beads, and tubes were removed from the rack to allow DNA elution. Tubes were once again set onto the magnets, and the clarified solution (containing DNA) was collected.

DNA was quantified using the Quant-iT PicoGreen dsDNA assay kit (Thermo Fisher Scientific, cat# P11496). 1 µL DNA sample was added to 14 µL TE diluted PicoGreen in MicroAmp optical 96-well plates (Thermo Fisher Scientific, cat# N8010560) as per the manufacturer's directions, sealed, and placed into a QuantStudio qPCR machine for analysis. Samples with DNA content greater than the target quantity of 25 ng/µL were diluted with MilliQ.

### Sheep blood DNA source

DNA methylation was analyzed in DNA extracted from the blood of 153 South Australian Merino sheep samples (80 transgenic Huntington's disease model sheep [OVT73 line] [*Jacobsen et al., 2010*] and 73 age-matched controls) aged from 2.9 to 7.0 years (*Figure 1A*). All protocols involving OVT73 sheep were approved by the Primary Industries and Regions South Australia (PIRSA, approval number 19/02) Animal Ethics Committee with oversight from the University of Auckland Animal Ethics Committee. The epigenetic age of the transgenic sheep carrying the *HTT* gene was not significantly different from controls (p=0.30, Mann–Whitney U test); therefore, the data derived from these animals was subsequently treated as one dataset (*Figure 1—figure supplement 1C*).

300 µL thawed blood samples were treated with two rounds of red cell lysis buffer (300 mM sucrose, 5 mM MgCl$_2$, 10 mM Tris pH8, 1% Triton X-100) for 10 min on ice, 10 min centrifugation at 1800 RCF, and supernatant removed between each buffer treatment. The resulting cell pellet was incubated in cell digestion buffer (2.4 mM EDTA, 75 mM NaCl, 0.5% SDS) and Proteinase K (500 µg/mL) at 50℃ for 2 h. Phenol:chloroform:isoamyl alcohol (PCI, 25:24:1; pH 8) was added at equal volumes, mixed by inversion, and placed in the centrifuge for 5 min at 14,000 RPM at room temperature (repeated if necessary). The supernatant was collected and combined with 100% ethanol at 2× volume, allowing precipitation of DNA. Ethanol was removed and evaporated, and 50 µL TE buffer (pH 8) was added to resuspend genomic DNA. DNA sample concentration was initially quantified using a nanodrop, followed by Qubit.

### Data processing and clock construction

A custom Illumina methylation array ('HorvathMammalMethyl40') was used to measure DNA methylation. These arrays include 36k CpG sites conserved across mammalian species, though not all probes are expected to map to every species (*Arneson et al., 2021*). Using QuasR (*Gaidatzis et al., 2015*), 33,136 probes were assigned genomic coordinates for sheep genome assembly OviAri4.

Raw .idat files were processed using the *minfi* package for RStudio (v3.6.0) with *noob* background correction (*Aryee et al., 2014*; *Triche et al., 2013*). This generates normalized beta values that represent the methylation levels at probes on a scale between 0 (completely unmethylated) and 1 (fully methylated).

185 CpG sites were selected for the sheep epigenetic clock by elastic net regression using the RStudio package *glmnet* (*Friedman et al., 2009*). The elastic net is a penalized regression model that combines aspects of both ridge and lasso regression to select a subset of CpGs that are most predictive of chronological age. 88 and 97 of these sites correlated positively and negative with age, respectively. Epigenetic age acceleration was defined as residual resulting from regressing DNAm age on chronological age. By definition, epigenetic age acceleration has zero correlation with chronological age. Statistical significance of the difference in epigenetic age acceleration between each male group (castrated versus intact) was determined using a non-parametric two-tailed Mann–Whitney U test applied to sexually mature sheep only (>18 months of age).

The human and sheep dual-species clock was created by combining our sheep blood and ear sourced data with human methylation data previously measured using the same methylation array ('HorvathMammalMethyl40') (*Arneson et al., 2021*). This data comprises 1848 human samples aged 0–93 years and includes 16 different tissues. The clock was constructed identically to the sheep only clock, with an additional age parameter *relative age* defined as the ratio of chronological age by maximum age for the respective species. The maximum age for sheep and humans was set at 22.8 years and 122.5 years, respectively, as defined in the anAge database (*de Magalhães et al., 2009*).

### Identification of age-associated and androgen-sensitive DMPs

Age-associated DMPs were identified using the weighted gene co-expression network analysis (WGCNA) function *standardScreeningNumericTrait* (*Langfelder and Horvath, 2008*), which calculates the correlation between probe methylation and age. WGCNA outputs Pearson correlation as standard; however, we repeated the analysis using Spearman correlation and good consistency was seen generally, with regions of interest (e.g., PAX6, FGF8, PAX5, HOXC4, and IGF1) showing very similar results. Significance values reported are uncorrected for multiple testing. Where mapped, gene names for the top 500 positively correlated probes were input into DAVID (*Dennis et al., 2003*) for functional classification analysis with the *O. aries* genes present on the methylation array as background. asDMPs were identified using a t-test of the difference between the slopes of linear regression lines applied to methylation levels across age in each sex. A difference in slope indicates that there is an interaction between age and sex for methylation status of a particular probe.

### TF binding analysis

TF binding at specific asDMPs was evaluated by entering the equivalent human probe position into the *interval search* function of the Cistrome Data Browser Toolkit, an extensive online collection of ChIP-seq data (*Mei et al., 2017*; *Zheng et al., 2019*). TFs binding CpGs of interest in human were analyzed using a custom R script, with ChIP-seq tracks being viewed by Cistrome link in the UCSC

genome browser alongside additional annotation tracks of interest for export and figure creation in Inkscape. To model the background levels of TF binding, 1000 replicates of 50 random probes sites were run in a similar manner using the Cistrome Human Factor dataset and compared to sheep orthologs in batches of 50 asDMPs using BEDtools (*Quinlan and Hall, 2010*) as well as custom R and Python scripts.

## Acknowledgements

We thank Ari Samaranayaka for his guidance with portions of the statistical analyses.

## Additional information

### Competing interests

Donna M Bond: Director and shareholder of Totovision, a small agricultural consultancy. Reuben R Hore: Commercial sheep famer. Timothy A Hore: is a shareholder and director of Totovision Ltd, a small agricultural and biotechnology consultancy. Steve Horvath: is a founder of the non-profit Epigenetic Clock Development Foundation which plans to license several patents from his employer UC Regents, including a patent for the mammalian assay utilised in this study (WO2020150705). The other authors declare that no competing interests exist.

### Funding

| Funder | Author |
| --- | --- |
| Paul G. Allen Frontiers Group | Steve Horvath |
| University of Otago | Timothy A Hore |

The funders had no role in study design, data collection and interpretation, or the decision to submit the work for publication.

### Author contributions

Victoria J Sugrue, Data curation, Software, Formal analysis, Validation, Investigation, Visualization, Methodology, Writing - original draft, Project administration, Writing - review and editing; Joseph Alan Zoller, Software, Formal analysis, Visualization, Methodology; Pritika Narayan, Data curation, Writing - review and editing; Ake T Lu, Formal analysis, Writing - review and editing; Oscar J Ortega-Recalde, Software, Formal analysis, Writing - review and editing; Matthew J Grant, C Simon Bawden, Skye R Rudiger, Donna M Bond, Reuben R Hore, Karen E Sears, Nan Wang, Xiangdong William Yang, Resources; Amin Haghani, Data curation, Software, Formal analysis, Validation, Investigation, Visualization, Methodology, Writing - review and editing; Michael Garratt, Formal analysis, Investigation, Writing - review and editing; Russell G Snell, Resources, Data curation, Writing - review and editing; Timothy A Hore, Conceptualization, Resources, Data curation, Formal analysis, Supervision, Funding acquisition, Validation, Investigation, Methodology, Project administration, Writing - review and editing; Steve Horvath, Conceptualization, Resources, Data curation, Software, Formal analysis, Supervision, Funding acquisition, Validation, Investigation, Methodology, Project administration, Writing - review and editing

### Author ORCIDs

Victoria J Sugrue  https://orcid.org/0000-0002-0427-5146
Joseph Alan Zoller  http://orcid.org/0000-0001-6309-0291
Amin Haghani  http://orcid.org/0000-0002-6052-8793
Karen E Sears  http://orcid.org/0000-0001-9744-9602
Xiangdong William Yang  http://orcid.org/0000-0003-3705-7935
Timothy A Hore  https://orcid.org/0000-0002-6735-225X

## Ethics

Animal experimentation: All protocols involving OVT73 sheep were approved by the Primary Industries and Regions South Australia (PIRSA, Approval number 19/02) Animal Ethics Committee with oversight from the University of Auckland Animal Ethics Committee. Sheep ear tissue was obtained from females and both intact and castrated male Merino sheep during routine on-farm ear tagging procedures in Central Otago, New Zealand. As a small piece of tissue is removed during the ear tagging process that is usually discarded by the farmer, we were able to source tissue and record the year of birth without altering animal experience, in accordance with the New Zealand Animal Welfare Act (1999) and the National Animal Ethics Advisory Committee (NAEAC) Occasional Paper No 2 (Carsons, 1998).

## Decision letter and Author response

Decision letter https://doi.org/10.7554/eLife.64932.sa1
Author response https://doi.org/10.7554/eLife.64932.sa2

# Additional files

## Supplementary files

• Supplementary file 1. Pearson correlation between age and methylation for all sheep.

• Supplementary file 2. Gene Ontology output for the top 500 probes positively associated with age.

• Supplementary file 3. Androgen-sensitive differentially methylated probes.

• Transparent reporting form

## Data availability

Data and code is available via GitHub https://github.com/VictoriaSugrue/sheepclock (copy archived at https://archive.softwareheritage.org/swh:1:rev:b7e3088e28025ac34513778ccb23368246fc532c).

The following dataset was generated:

| Author(s) | Year | Dataset title | Dataset URL | Database and Identifier |
|---|---|---|---|---|
| Sugrue VJ, Zoller JA, Narayan P, Lu AT, Ortega-Recalde OJ, Grant MJ, Bawden CS, Rudiger SR, Haghani A, Bond DM, Garratt M, Sears KE, Wang N, Yang XW, Snell RG, Hore TA, Horvath S | 2021 | Castration delays epigenetic aging and feminises DNA methylation at androgen-regulated loci | https://github.com/VictoriaSugrue/sheepclock | GitHub, sheepclock |

The following previously published dataset was used:

| Author(s) | Year | Dataset title | Dataset URL | Database and Identifier |
|---|---|---|---|---|
| Horvath S, Snell RG | 2020 | DNA methylation profiles from a transgenic sheep model of Huntington's disease | https://www.ncbi.nlm.nih.gov/geo/query/acc.cgi?acc=GSE147003 | NCBI Gene Expression Omnibus, GSE147003 |

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
