## [Decision Letter]

**Acceptance summary:**

The paper by Sugrue et al. demonstrates that androgen deficiency (induced by castration) in male sheep delays epigenetic aging. The paper also includes examples from human and bat samples, and provides new knowledge on sexual dimorphism and aging that should be of interest to everyone in the epigenetics field.

**Decision letter after peer review:**

Thank you for submitting your article "Castration delays epigenetic aging and feminises DNA methylation at androgen-regulated loci" for consideration by *eLife*. Your article has been reviewed by 2 peer reviewers, including Sara Hägg as the Reviewing Editor and Reviewer #1, and the evaluation has been overseen by Matt Kaeberlein as the Senior Editor. The following individual involved in review of your submission has agreed to reveal their identity: Amr Sawalha (Reviewer #3).

The reviewers have discussed the reviews with one another and the Reviewing Editor has drafted this decision to help you prepare a revised submission.

Summary:

The paper by Sugrue et al. describes an epigenetic clock in sheep and explores how castration affects epigenetic aging. The study demonstrates that androgen deficiency (induced by castration) in male sheep delays epigenetic aging. Androgen specificity of probes that show differences in age-dependent methylation changes between castrated and non-castrated males was inferred by enrichment in AR binding sites suggesting presumed regulation by androgens.

Essential revisions:

1. The aim of the study is not adequately described. There are incentives for the epigenetic clock creation, but the results also include age-related changes on the methylome?

2. Not the same order of presentation in methods and results. Age-associated changes in the methylome comes first in results but not in methods.

3. Any batch effects corrected for in the methylation data?

4. Pearson correlations assumes linearity, sensitivity analysis using Spearman correlation may be needed?

5. Figure 1:

a. In the analysis correlating epigenetic changes with age, presented in Figure 1B, please clarify how these data were corrected for multiple testing.

b. Is there a difference between females and intact males in the analysis correlating methylation levels with chronological age (Figure 1B)?

c. The figure legend in does not reflect the data presented in the figure.

6. The epigenetic clock:

a. Is the castrated samples included in the elastic net regression model in the clock creation? This is unclear, and I don´t think these samples should be included as they deviate from normal aging in sheep, at least in 18 months old sheep.

b. Figure 2A. The correlation in sheep blood is not as good as in other selections. Why is this? There seems to be an overrepresentation of young sheep in the data, can this be corrected for using weighted effects in the elastic net to get a better prediction? Or what happens if the Huntington samples are removed? Are they causing this pattern?

c. Figure 2E seems to be identical to 2C, what is the difference here? Not adequately described.

d. The rationale for creating a dual species clock is not clear. What does it add?

e. Figure 3A, I don´t understand how the ranking of the samples was done? If you have age-matched castrated and intact male sheep, why don´t you also use a matched control setting for the test statistic to calculate the DNAmAge difference? That would most likely account for other unmeasured confounders as well.

f. Figure 3B, P-value in text and figure legend does not match.

7. I do not agree on this statement: "We found there was a sharp inflection in p value after approximately the 50 most significant probes (Figure S4A) and thus represented a natural cut-off for analysis." Looking into Table S3 and the p-values for the CpGs, this is not apparent. I suggest make a FDR-level cut-off and redo enrichment analyses as well.

8. The mouse methylation data are not described and the statistical test used in Figure 6 is not explained. Please put labels on top of Figure 6A and B.

9. It will be informative to examine DNA methylation differences between age-matched female and male sheep using a subset of the study samples. Also more specifically, is there a difference in methylation levels among the asDMPs identified?

10. To what extent does castration reverse methylation differences between males and females and/or methylation levels in asDMP compared to age-matched females? In other words do androgens completely or partly explain the difference between males and females?

11. It would have been nice if serum samples are available to explore the correlation between methylation levels in asDMP and androgen levels, but this might not be feasible.

12. The authors show significantly reduced mass of male lambs after castration. Within the castrated and intact groups separately, is there any correlation between mass and DNA methylation levels at any of the asDMP? This will help establish a "phenotypic" change associated with the DNA methylation changes induced by androgen deficiency.

13. Any specific positional annotation enrichment for age-correlated CpG sites and asDMP relative to CpG islands, shores, etc.

14. Can the authors further speculate about the reason behind tissue-specificity for asDMP (e.g. MKLN1), and how the presence or absence of androgens might be mechanistically linked to changes in DNA methylation? Why is there no difference seen in blood mammal tissues? The discussion around this needs to dig a little deeper.

---

## [Author Response]

Essential revisions:

1. The aim of the study is not adequately described. There are incentives for the epigenetic clock creation, but the results also include age-related changes on the methylome?

In response to the reviewers comments we have made more of an effort to introduce the need to explore the fundamentals of molecular aging through DNA methylation (and its association with age). Key passages edited include:

“Moreover, sheep are extensively farmed (and males castrated) in many countries, allowing incidental study of the effect of sex and sex hormones in aging to occur without increasing experimental animal use (Russell and Burch, 1959). Yet, exploration of the molecular aging process in sheep is relatively under developed, particularly from the perspective of epigenetics and sex.”

and

“Here we present the first sheep epigenetic clock and quantify its median error to 5.1 months, ~3.5-4.2 % of their expected lifespan. Validating the biological relevance of our sheep model, we find age associated methylation at genes well-characterised for their role in development and aging in a wide range of animal systems.”

2. Not the same order of presentation in methods and results. Age-associated changes in the methylome comes first in results but not in methods.

We have now placed the "Identification of age-associated and androgen-sensitive DMPs" section earlier in the methods, before description of the clock.

3. Any batch effects corrected for in the methylation data?

In knowing the risk of batch effects potentially skewing our data, we sought to eliminate them from our main test comparison (i.e. castrates vs intacts) by arranging them on our array chips alternately (i.e. castrate_1, intact_1, castrate_2, intact_2….etc). Having put these protections in place, it appears they were not needed – hierarchical clustering revealed some clustering on the basis of sex and age as indicated in the manuscript, however, did not reveal any structuring on batch.

We have since clarified this point in the manuscript.

“There was some sub-clustering based on sex and age; however, there was no separation based on known underlying pedigree variation or processing batches”.

4. Pearson correlations assumes linearity, sensitivity analysis using Spearman correlation may be needed?

In response to the reviewers’ comment, we compared correlation between chronological age and methylation levels at each CpG using Pearson and Spearman, and found good consistency between the methods for all samples and for Ewe ear punch only (see Author response image 1 and Author response image 2). Importantly, sites we picked out as being biologically interesting showed high correlation using either Pearson or Spearman (highlighted red).

**Author response image 2. respfig2:** 

While we are pleased to have done this analysis, we are not convinced the results warrant inclusion in the manuscript. Nevertheless, details of what was done have now been included in the methods section.

“WGCNA outputs Pearson correlation as standard; however, we repeated the analysis using Spearman correlation and good consistency was seen generally, with regions of interest (e.g. PAX6, FGF8, PAX5, HOXC4 and IGF1) showing very similar results.”

5. Figure 1:a. In the analysis correlating epigenetic changes with age, presented in Figure 1B, please clarify how these data were corrected for multiple testing.

No correction for multiple testing was done for this figure (i.e. only uncorrected p-values are showing). Nevertheless, the findings are still significant even after Bonferroni correction. This has now been clarified in the methods section.

b. Is there a difference between females and intact males in the analysis correlating methylation levels with chronological age (Figure 1B)?

In response to this comment we are now showing stratified data for tissue and sex in Figure 1—figure supplement 2. We found that sex and castration status had much less effect on age associations compared when tissue was used to stratify the data. This has been updated in the text as follows:

“When we stratified our data, we found considerable differences in age association with regard to tissue of origin (Figure 1—figure supplement 2), however, this is consistent with other clock studies. Sex and castration status also produced group-specific age association hits, however, this was comparatively less than for tissue, with many of the most significantly associated sites being shared between females and males (blood, 45 shared sites) and females, intact males and castrated females (ear, 84 shared sites) (Figure 1—figure supplement 2B-C).”

c. The figure legend in does not reflect the data presented in the figure.

We agree, this was an oversight – "Creation of the epigenetic clock in sheep." refers specifically to just one part of the Figure 1A, not the whole figure. As such, we have changed this figure title to "Association between age and DNA methylation in sheep".

6. The epigenetic clock:a. Is the castrated samples included in the elastic net regression model in the clock creation? This is unclear, and I don´t think these samples should be included as they deviate from normal aging in sheep, at least in 18 months old sheep.

Castrated sheep were included in the elastic net regression as indicated in Figure 1. We felt this was important to include them in the clock, as modelling epigenetic age in castrates forms a central part of our analysis (if the castrates were not included, we risk having a clock that held sites not suitable for their population).

b. Figure 2A. The correlation in sheep blood is not as good as in other selections. Why is this? There seems to be an overrepresentation of young sheep in the data, can this be corrected for using weighted effects in the elastic net to get a better prediction? Or what happens if the Huntington samples are removed? Are they causing this pattern?

We feel there are two potential factors driving the lower correlation in blood. Firstly, there was far fewer sheep included in the blood only set, and as pointed out by the reviewers, an overrepresentation of young sheep. This means the clock will be less predictive in general, particularly at advanced ages. Secondly, blood is likely more heterogeneous as a tissue compared to ear punches.

There was no significant difference in predicted epigenetic age for Huntington sheep compared to controls (p=0.3), therefore, we think this is an unlikely contributing factor.

To clarify this point in the manuscript, we have added the following passage to the Results section:

“When blood and ear clocks are constructed separately, the ear clock outperformed the blood clock in the LOOCV analysis (0.97 vs 0.75 correlation respectively, Figure 2B-C). While this may be related to blood being a more heterogeneous tissue than ear punch, less samples and an overrepresentation of young individuals is a likely driver of this effect.”

Further, to assist readability, we have changed the order of Figure 2A-C so that the multi-tissue clock (that was used for the majority of the analysis) is presented first (Figure2A), with the blood and skin clocks following (Figure 2B and 2C respectively)

c. Figure 2E seems to be identical to 2C, what is the difference here? Not adequately described.

Figure 2E represents a sheep and human multi-tissue clock that has been applied to only sheep, whereas Figure 2C (in the old manuscript) is a sheep-only multi-tissue clock applied to sheep only. We have clarified this in the Figure legend as follows.

**“**Figure 2. Comparison of chronological age (x-axis) and epigenetic age (y-axis) for a variety of clock models trained on (A-C) sheep only, or (D-G) humans and sheep. […] Maximum lifespan values used were for human and sheep respectively were 122.5 years and 22.8 years. Each data point represents one sample, coloured based on origin.”

d. The rationale for creating a dual species clock is not clear. What does it add?

In addition to defining a single set of sites that act as a clock in both species, a dual-species clock provides an extra 'sanity-check' on our data because spurious age associations will be reduced by having an additional species and extra samples. Indeed, the demonstration that epigenetic age deceleration of castrates was found using the dual-species clock provides additional support for the original observation using the sheep clock alone.

This has now been made clearer in the Results section:

“In order to broaden applicability of the sheep clock, and also substantiate its general characteristics, two human and sheep dual-species clocks were also constructed.”

and

“Notably, the age deceleration observed in castrates was corroborated using the human and sheep dual-species clock (Figure 3—figure supplement 3, p=0.04), thus providing additional support for this finding.”

e. Figure 3A, I don´t understand how the ranking of the samples was done? If you have age-matched castrated and intact male sheep, why don´t you also use a matched control setting for the test statistic to calculate the DNAmAge difference? That would most likely account for other unmeasured confounders as well.

The 'ranking' of the samples is done by epigenetic age, and only relates to Figure 3A (note, statistics was not applied on this ranked data). The reason we presented Figure 3A in this manner was so that it was obvious to the reader that as the sheep aged (chronologically), divergence in epigenetic age between castrated and intact males became greater.

f. Figure 3B, P-value in text and figure legend does not match.

This was a typographical error on our behalf – The "0.01" in the legend of Figure 3B was missing a digit (i.e. 0.018). This has now been corrected.

7. I do not agree on this statement: "We found there was a sharp inflection in p value after approximately the 50 most significant probes (Figure S4A) and thus represented a natural cut-off for analysis." Looking into Table S3 and the p-values for the CpGs, this is not apparent. I suggest make a FDR-level cut-off and redo enrichment analyses as well.

While we do believe there is an inflection point in p-value around this cut-off, we have nevertheless removed the top 50 as a cut-off for subsequent analysis; and have instead performed cistrome enrichments for all asDMPs identified. As this extended analysis required automation and updating of the cistrome database output, the numbers in Figure 5 have changed slightly, however, the interpretation remains the same.

Importantly, this extended analysis revealed that androgen receptor (AR) and the glucocorticoid receptor (NR3C1) have enriched binding in only the top ~200 most significant asDMPs, before returning to the expected background. This analysis is now included in Figure 4—figure supplement 4.

8. The mouse methylation data are not described and the statistical test used in Figure 6 is not explained. Please put labels on top of Figure 6A and B.

Figure 6 has been completely reconstructed on account of exciting new data. Nevertheless, we have labelled tissue and species much more clearly and described the statistical test used in the same manner as Figure 4.

9. It will be informative to examine DNA methylation differences between age-matched female and male sheep using a subset of the study samples. Also more specifically, is there a difference in methylation levels among the asDMPs identified?

We do not understand what the reviewers are asking for in this question – age matched female and male sheep methylation for the asDMPs is shown in Figure 4.

10. To what extent does castration reverse methylation differences between males and females and/or methylation levels in asDMP compared to age-matched females? In other words do androgens completely or partly explain the difference between males and females?

As in Figure 4, for the iconic asDMPs (MKLN1, ETAA1, *LMO4*, KIAA2026), androgens almost completely explain the differences between males and females. Despite being a strong effect at this site, there are some sexually dimorphic probes that are androgen-insensitive, so the effect is partial.

We have altered the discussion to clarify this point:

“Comparison of intact and castrated males also allowed us to identify several age-related DMPs that display clear androgen-sensitivity (asDMPs, Figure 4A-D), with castrated males exhibiting a feminised methylation profile compared to intact counterparts at these sites”.

11. It would have been nice if serum samples are available to explore the correlation between methylation levels in asDMP and androgen levels, but this might not be feasible.

Unfortunately this is not feasible (blood/serum was not taken from the rams where ear punches were also taken). While beyond the scope of the current paper, we agree this will be an interesting experiment to perform in the future – there is reasonable variation in ram testosterone level (Clarke et al., 2012; https://doi.org/10.1210/en.2011-1634), and this may be reflected in asDMP methylation. Nevertheless, we have performed live weight comparisons to methylation (as suggested by reviewers below) which somewhat acts as a proxy for testosterone level.

12. The authors show significantly reduced mass of male lambs after castration. Within the castrated and intact groups separately, is there any correlation between mass and DNA methylation levels at any of the asDMP? This will help establish a "phenotypic" change associated with the DNA methylation changes induced by androgen deficiency.

This was a really helpful suggestion that has given us new insight into the paper.

We examined the relationship between mass in the young males (i.e. <1 year old, the only sheep with mass recorded in this project) and DNA methylation and asDMPs. We found that, when individual sites were compared, the relationship to mass was on its own largely insignificant. For example, our 4 'iconic' asDMPs is given in Author response image 3 – of the 8 correlations shown here, only one was deemed significant (ram mass vs methylation at KIAA2026; p=0.0121). Nevertheless, it appeared that in general, there was a negative relationship between mass and methylation in intact males (green). For the castrates (blue), methylation at asDMPs and mass had a positive relationship.

**Author response image 3. respfig3:** 

We then set out to explore how robust these associations were by comparing groups of 100 asDMPs at a time, and asking, (1) is there a difference in association of mass and asDMP methylation that changes for castrates and intact males, that is dependent upon asDMP prominence? and (2) does the mass vs methylation associations differ significantly between intact males and castrates? Indeed, starting with the 100 most prominent asDMPs identified, average association between mass and methylation was negative for intact rams and positive for castrates (p=1x10-9). To be sure these correlations were biologically meaningful, we sampled through less prominent asDMPs in a sliding window of 100, finding that significance merged with that of randomly chosen 100 CpG sites (10000 replicates) after the top ~1000 asDMPs.

What this analysis shows is that divergence of methylation at asDMP sites occurs early (i.e. as soon as physical traits like mass become dimorphic between intact and castrated males), and is detectable even before differences in epigenetic age are manifested. To illustrate this point, we have now included new panels in Figure 4—figure supplement 6, that show (B) the associations between mass and asDMP methylation in young sheep and (panel C) where significant differences are lost. The manuscript text has also been updated accordingly.

13. Any specific positional annotation enrichment for age-correlated CpG sites and asDMP relative to CpG islands, shores, etc.

The most striking positional enrichments found in this study was between asDMPs and the hormone related transcription factors (e.g. AR and NR3C1; as reported in Figure 5). We did perform CGI enrichment analysis – for the asDMPs, which predominantly demethylate with age in males, overlap with CGIs was observed at almost exactly the same rate as CpGs that are negatively correlated with age (i.e. 2.1% and 2.5% respectively). In contrast, sites where methylation is positively correlated with age (i.e. 'hypermethylated' over time) have lower rates of CGIs (0.32%). While it is useful to have verified that CGI enrichment is similar between asDMPs and similarly demethylating age-associated sites, we do not feel this warrants discussion in the manuscript.

14. Can the authors further speculate about the reason behind tissue-specificity for asDMP (e.g. MKLN1), and how the presence or absence of androgens might be mechanistically linked to changes in DNA methylation? Why is there no difference seen in blood mammal tissues? The discussion around this needs to dig a little deeper.

Again, this was a very helpful comment – we took advantage of new data from the pan-mammalian methylation array in mouse and classified them on the basis of whether there was any sex differences in methylation during aging. For MKLN1 we found mouse tail, kidney and muscle all showing asDMP behaviour, as in sheep ear punch. In contrast, mouse blood, cerebellum, cortex, heart, liver, spleen, striatum, and whole brain all lacked androgen sensitivity at this site (see more in revised Figure 6). Interestingly, those mouse tissues showing androgen sensitive methylation changes had significantly higher expression of the AR compared to tissue that do not. In addition to completely reformulating Figure 6, we have identified sex-dependent methylation changes in bats from published data of others (Figure 6—figure supplement 7). The results and discussion have been updated to reflect new data – we think together this represents an important new clues between TF binding and age-association, as well as uncovers some broader principles likely to be shared across divergent mammalian groups.